# Comparison of the predictive ability for perinatal acidemia in neonates between the NICHD 3-tier FHR system combined with clinical risk factors and the fetal reserve index

**Ninlapa Pruksanusak[1], Natthicha Chainarong[1] \*, Siriwan Boripan[1], Alan Geater[2]**

**1** Department of Obstetrics and Gynecology, Faculty of Medicine, Prince of Songkla University, Hat Yai, Songkhla, Thailand, **2** Epidemiology Unit, Faculty of Medicine, Prince of Songkla University, Hat Yai, Songkhla, Thailand

\* chnatthicha@gmail.com

## Abstract

### Background

Electronic fetal monitoring alone is a poor screening test for detecting fetuses at risk of acidemia or asphyxia. We aimed to evaluation of predictive ability of the National Institute of Child Health and Human Development (NICHD) 3-tier fetal heart rate (FHR) system combined with the maternal, obstetric, and fetal risk factors for predicting perinatal acidemia, and to compare this with the predictive of the NICHD 3-tier system alone, and the Fetal Reserve Index (FRI).

### Methods

A retrospective cohort study was conducted among singleton term pregnant women. Fetal heart rate tracings of the last two hours before delivery were interpreted into the NICHD 3-tier FHR classification system by two experienced obstetricians. Demographic data were compared using the $\chi 2$ or Fisher's exact test for categorical variables and the Student's t test for continuous variables. Logistic regression model was used to identify factors associated with perinatal acidemia in neonates. The Odds ratios (OR) and probabilities with 95% confidence intervals (CI) were calculated.

### Results

A total of 674 pregnant women were enrolled in this study. Using the NICHD 3-tier FHR categories I and II combined with the selected risk factors (AUC 0.62) had a better performance for perinatal acidemia prediction than the NICHD 3-tier FHR alone (AUC 0.55) and the FRI (AUC 0.52), (P<0.01). Improvement of predicting perinatal acidemia was found when NICHD category I was combined with preeclampsia or arrest disorders of labor (OR 3.2, 95% CI 1.30–7.82) or combined with abnormal second stage of labor (OR 6.19, 95% CI 1.07–36.06) and when NICHD category II was combined with meconium-stained amniotic fluid (OR 4.73, 95% CI 2.17–10.31).

**Data Availability Statement:** All relevant data are within the paper and its Supporting Information files.

**Funding:** This work was supported by Faculty of Medicine, Prince of Songkla University, Songkhla, Thailand. The funder had no role in the study design, data collection and analysis, interpretation of data, writing of the report, and the decision to submit the article for publication.

**Competing interests:** The authors have declared that no competing interests exist.

## Conclusions

The NICHD 3-tier FHR categories I or II combined with selected risk factors can improve the predictive ability of perinatal acidemia in neonates compared with the NICHD 3-tier system alone or the FRI.

## Introduction

Electronic fetal monitoring (EFM) is one of the most widely used tools to monitor fetal health status. Experts have created intrapartum fetal heart rate (FHR) interpretation systems, including the National Institute of Child Health and Human Development (NICHD) 3-tier FHR classification system, the 5-tier color-coded graded FHR, and the 3-tier FHR classification system of the International Federation of Gynecology and Obstetrics (FIGO) [1–3]. These systems were developed to improve the predictive ability for identifying fetuses at risk of abnormal acid-base status [1–3], a condition which leads to an increased risk of cerebral palsy [4], so timely management can be undertaken before fetal decompensation. Earlier studies have shown that EFM alone is a poor screening test to detect abnormal acid-base status and reduce fetal death or cerebral palsy. EFM also contributes to increased cesarean delivery rates because of its high false positive rate. The lack of a quantifiable relationship between an EFM pattern and significant fetal acidemia leading to an increased probability of cerebral palsy is a point of concern [5–11].

Using the intrapartum FHR pattern combined with maternal and fetal associated risk factors may improve the prediction of fetuses at risk of perinatal acidemia or asphyxia [9, 12–14]. The fetal reserve index (FRI) proposed by Evans et al. is a scoring system that combines the EFM parameters (FHR, variability, accelerations, decelerations) and uterine activity with maternal, obstetrical, and fetal risk factors. Each of eight categories is assigned a "1" if normal and a "0" if abnormal or in the presence of risk factors. All eight categories being classified as normal would result in a score of 100 (8/8). An abnormal FRI is defined by a score ≤25 (2/8). The FRI is more accurate in predicting cerebral palsy than using the NICHD 3-tier FHR category III alone [9, 13, 14]. However, an important limitation of the FRI is that it requires experienced interpreters who can meticulously analyze FHR patterns, which can dynamically change during labor, while, NICHD 3-tier FHR systems have been used widely in general practice for many years and are more user-friendly for most healthcare providers in obstetrics.

Therefore, to try to optimize the use of these available tools, we combined the maternal, obstetric, and fetal risk factors with the NICHD 3-tier FHR system to evaluate the predictive ability for perinatal acidemia and compared this combined technique with the NICHD 3-tier system alone and the FRI.

## Materials and methods

A retrospective cohort study was conducted at Songklanagarind Hospital, a tertiary center in southern Thailand, between March and December 2015 after approval was granted by the Human Research Ethics Committee of the Faculty of Medicine, Prince of Songkla University. The need for informed consent was waived because of the retrospective study design. All singleton pregnancies having labor at a gestational age of ≥ 37 weeks who had fetal monitoring for the two hours before delivery and umbilical cord blood gas results were included. The FHR tracings were interpreted and managed according to the NICHD 3-tier FHR system [1, 7, 15].

Subjects excluded from the study were women who (1) underwent a cesarean section prior to labor, or (2) had suspected fetal anomalies. The required study population size was determined based on comparison of area under the curve (AUC) of receiver operating characteristic (ROC) curves between the NICHD 3-tier system alone (estimated at 0.55) and the NICHD 3-tier system combined with risk factors or the FRI (estimated at 0.70) with approximately 12% of neonates having acidemia. For power of 80% and alpha of 0.05, and allowing for up to 20% incompleted data, 658 were required [16].

The primary outcome was evaluation of the predictive ability of the NICHD 3-tier system combined with the maternal, obstetric, and fetal risk factors to predict perinatal acidemia in neonates, and to compare this with the predictive of the NICHD 3-tier system alone, and the FRI.

We reviewed the medical records extracted from the computerized hospital database system. Maternal, obstetric, and fetal risk factors based on the FRI recommendations were recorded [9]. Briefly, the maternal variables were chronic medical disorders (e.g., cardiac, respiratory, endocrine), gestational hypertension, maternal age, body mass index, smoking, and weight gain. Obstetric variables were parity, gestational age, fetal growth, amniotic fluid, placental and umbilical abnormalities, use of oxytocin, bleeding events, rupture of membrane, duration of labor, and route of delivery. Fetal variables were abnormal Doppler study, evidence of genetic disorders/infection, amnioinfusion, and meconium passage [9]. Neonatal outcomes included standard outcomes such as birth weight; Apgar scores at 1, 5, and 10 minutes; umbilical cord blood gases; cardiac, respiratory, and neurological events; neonatal resuscitations; and intensive care unit admission. Perinatal acidemia was defined as umbilical pH $\leq$7.2 or base deficit $\geq$12 mmol/L within 60 minutes after birth [4].

The FHR tracings were independently reviewed by two experienced obstetricians. One reviewer (SB) was a final-year maternal-fetal medicine (MFM) fellow and the other reviewer (NC) was an MFM board-certified practitioner. Each reviewer interpreted the tracings following the NICHD 3-tier FHR classification system and was blind to the associated clinical data [1]. The kappa coefficient was used to assess interobserver variability for EFM parameters such as NICHD category (I, I, III), FHR (bradycardia, normal, tachycardia), accelerations and decelerations (presence or absence), variability (absent/decrease, normal, increase). For contractions, abnormal uterine activity was defined as > 8 contractions in 20 minutes based on the FRI recommendations [9].

Demographic data were compared using $\chi^2$ or Fisher's exact test, where appropriate, for categorical variables and the Student's $t$ test for continuous variables. The Odds ratios (OR) and likelihood ratios (LR) with 95% confidence intervals (CI) were calculated from the regression coefficients. Receiver operating characteristic (ROC) curves were created to compare the abilities of the NICHD 3-tier system combined with the significant risk factors, the NICHD 3-tier system alone, and the FRI to predict perinatal acidemia neonates by area under the curve (AUC). Results with a $P$ value < .05 were considered statistically significant. The statistical analyses were performed using STATA software version 14.2 (StataCorp, College Station, TX, USA) and R program version 4.0.5.

## Results

During the study period, 674 term pregnancies were reviewed. Eighty neonates (11.8%) had evidence of perinatal acidemia. Of these, 60 neonates had pH $\leq$ 7.2, 47 neonates had base deficit $\geq$ 12 mmol/L, and 27 neonates had both pH $\leq$ 7.2 and BE $\geq$ 12 mmol/L. The baseline characteristics, antepartum and intrapartum variables, and neonatal outcomes were compared between women who delivered neonates with and without evidence of perinatal acidemia

**Table 1. Baseline characteristics, antepartum and intrapartum variables, and neonatal outcomes.**

| | Acidemia (n = 80) | Non-acidemia (n = 594) | P |
|---|---|---|---|
| Advanced maternal age | 15 (18.8) | 122 (20.5) | 0.82 |
| Median gestational age at delivery (weeks) (IQR) | 39 (38,40) | 39 (38,40) | 0.14 |
| BMI $\geq$ 30 (kg/m$^2$) | 4 (5) | 32 (5.4) | 0.76 |
| Prior cesarean section | 2 (2.5) | 42 (7.1) | 0.19 |
| Preeclampsia | 4 (5) | 12 (2) | 0.11 |
| Fetal growth restriction | 0 | 3 (0.5) | 1 |
| Macrosomia | 1 (1.2) | 3 (0.5) | 0.39 |
| Oligohydramnios | 3 (3.8) | 8 (1.3) | 0.13 |
| Polyhydramnios | 0 | 1 (0.2) | 1 |
| Rupture of membrane | 69 (86.2) | 475 (80) | 0.23 |
| Arrest disorders of labor | 11 (13.8) | 48 (8.1) | 0.14 |
| Meconium passage | 18 (22.5) | 63 (10.6) | 0.004 |
| Chorioamnionitis | 3 (3.8) | 5 (0.8) | 0.05 |
| Abnormal second stage of labor | 2 (2.5) | 12 (2) | 0.67 |
| **Neonatal outcomes** | | | |
| Median birth weight (grams) (IQR) | 3138 (2892,3409) | 3072 (2841,3390) | 0.41 |
| Median umbilical pH (IQR) | 7.17 (7.12,7.21) | 7.32 (7.29,7.36) | < 0.001 |
| Median umbilical base deficit (mmol/L) (IQR) | 12.35 (9.90,14.4) | 5.60 (3.80,7.60) | < 0.001 |
| Umbilical pH $\leq$ 7.2 | 60 (75) | 0 | < 0.001 |
| Umbilical base deficit $\geq$ 12 mmol/L | 47 (58.7) | 0 | < 0.001 |
| 1-minute Apgar score < 7 | 36 (45) | 24 (4) | < 0.001 |
| Intensive neonatal resuscitation | 23 (28.7) | 15 (2.5) | < 0.001 |
| Intubation | 3 (3.8) | 5 (0.8) | < 0.001 |
| NICU admission | 4 (5) | 9 (1.5) | < 0.001 |

Data are given as n (%) unless otherwise specified.

IQR, interquartile range; BMI, body mass index; NICU, neonatal intensive care unit.

(Table 1). Women who delivered neonates with evidence of perinatal acidemia had significantly higher rates of meconium-stained amniotic fluid than those without acidemia. No cases with epidural anesthesia were found in both groups.

Based on the FHR tracings, interobserver variability was very good for the presence of variable decelerations and pathologic decelerations (kappa 0.90–1.0); good for FHR variability, prolonged decelerations, NICHD category (kappa 0.64–0.72); and moderate for accelerations and presence of late decelerations (kappa 0.46–0.48). The EFM parameters according to the NICHD 3-tier FHR system were compared between the two groups (Table 2). More cases of pathologic decelerations and prolonged duration of abnormal tracing were observed in women who delivered neonates with evidence of perinatal acidemia than those without acidemia. However, no significant differences in NICHD categories I and II were found between the two groups. No cases of NICHD category III were found in our study.

Fig 1 demonstrates a comparison of the predictive ability of the NICHD 3-tier system combined with the maternal, obstetric, and fetal risk factors, the NICHD 3-tier system alone, and the FRI. The NICHD 3-tier system combined with significant risk factors had a better predictive ability for perinatal acidemia in neonates when compared with the others (P < .01). Our study showed that the NICHD 3-tier system alone and the FRI had a poor predictive abilities for perinatal acidemia in neonates with AUC values of 0.55 and 0.52, respectively. The power to predict perinatal acidemia was improved when NICHD category I was combined with

**Table 2. EFM parameters according to the NICHD 3-tier interpretation system.**

|  | Acidemia (n = 80) | Non-acidemia (n = 594) | *P* |
|---|---|---|---|
| Minimal variability | 10 (12.5) | 125 (21) | .1 |
| Marked variability | 1 (1.2) | 5 (0.8) | .53 |
| Pathologic deceleration | 43 (53.8) | 224 (37.7) | .008 |
| Variable deceleration | 36 (45) | 167 (28.1) | .27 |
| Late deceleration | 9 (11.3) | 46 (7.74) | 1 |
| Prolonged deceleration | 2 (2.5) | 38 (6.4) | .06 |
| Presence of tachycardia | 2 (2.5) | 5 (0.8) | .197 |
| Abnormal uterine activity | 0 | 2 (0.3) | 1 |
| NICHD category |  |  | .08 |
| Category I | 31 (38.8) | 295 (49.7) |  |
| Category II | 49 (61.2) | 299 (50.3) |  |
| Median time of abnormal tracing (minutes) (IQR) | 62.5 (36,112.8) | 47 (30,66) | .025 |

Data are given as n (%) unless otherwise specified.

NICHD, National Institute of Child Health and Human Development; IQR, interquartile range.

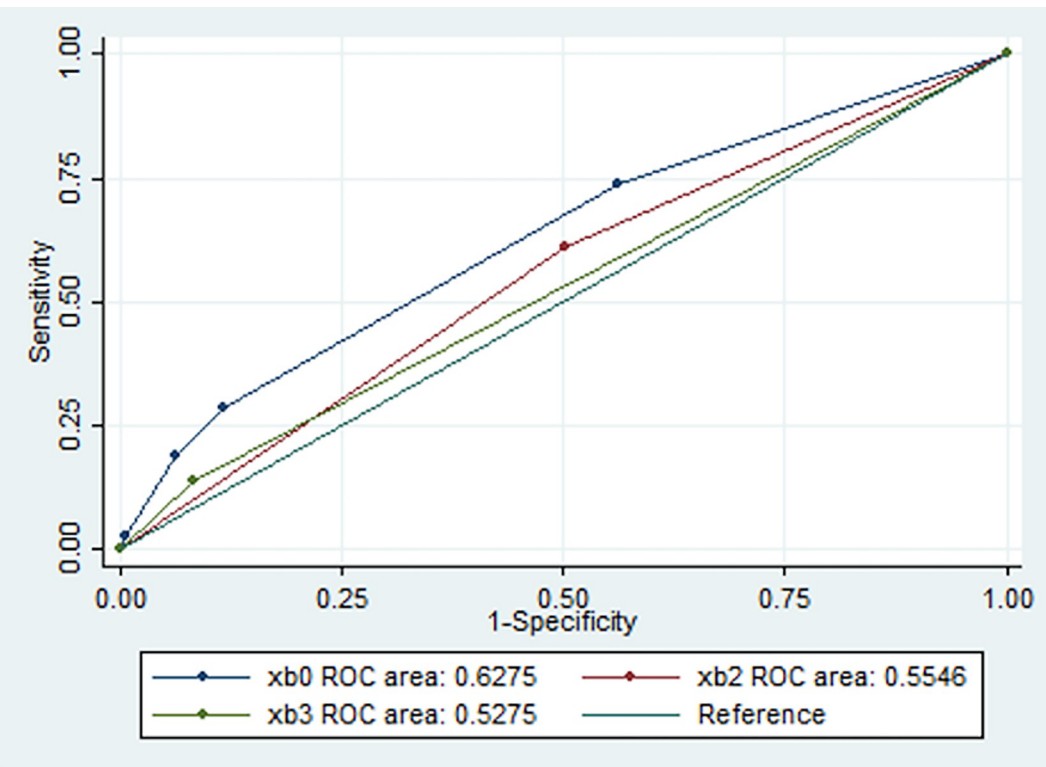

**Fig 1. Receiver operating curves (ROC) demonstrating the comparison of predictive abilities among the NICHD 3-tier system combined with significant risk factors, the NICHD 3-tier system alone, and the Fetal Reserve Index.** xb0, the NICHD 3-tier system combined with significant risk factors (preeclampsia, abnormal labor; arrest disorders of labor or abnormal second stage of labor, meconium-stained amniotic fluid); xb2, the NICHD 3-tier system alone; xb3, the Fetal Reserve Index. *p* value 0.01.

**Table 3. Predictive ability of NICHD 3-tier categories I and II alone and combined with significant clinical risk factors for perinatal acidemia.**

| NICHD Category | Adjusted OR (95% CI) | Probability (95% CI) | P |
|---|---|---|---|
| Category I | 1 (reference) | 0.075 (0.049–0.112) | < .001 |
| Category I with preeclampsia or arrest disorders of labor | 3.20 (1.304–7.823) | 0.208 (0.106–0.293) | |
| Category I with abnormal second stage of labor | 6.19 (1.079–36.063) | 0.33 (0.084–0.732) | |
| Category II | 1.68 (0.956–2.958) | 0.12 (0.088–0.161) | |
| Category II with meconium-stained amniotic fluid | 4.73 (2.173–10.312) | 0.277 (0.168–0.420) | |

preeclampsia or abnormal labor (arrest disorders or prolonged second stage of labor) and NICHD category II was combined with and meconium-stained amniotic fluid (Table 3).

## Discussion

The NICHD 3-tier system combined with significant risk factors of preeclampsia, abnormal labor (arrest disorders or prolonged second stage of labor), or meconium-stained amniotic fluid more accurately predicted perinatal acidemia in neonates than the NICHD 3-tier system alone or the FRI. Our study showed that NICHD categories I and II were better able to predict perinatal acidemia when NICHD category I was combined with preeclampsia or abnormal labor and when NICHD category II was combined with meconium-stained amniotic fluid than the NICHD 3-tier system alone.

We found that, even when classified as NICHD category I, some pregnant women delivered neonates with evidence of perinatal acidemia, as also found in previous studies [5, 17]. Our study also showed a better prediction of perinatal acidemia when using NICHD category I combined with certain specific risk factors (preeclampsia and abnormal labor). It is well known that preeclampsia is an important risk factor for birth asphyxia in neonates. Preeclampsia includes the failure of endovascular invasion by cytotrophoblasts that results in high-resistance, low-capacitance spiral arteries and predisposes to hypoperfusion, hypoxia, and reperfusion injury of the placenta [18–21]. This pre-existing compromising placental function accompanied with reduction in uteroplacental perfusion during repeated uterine contractions results in fetal hypoxia and acidosis [22].

The 2008 NICHD electronic fetal monitoring consensus guideline and previous studies reported that NICHD category I had a strong association with normal fetal acid-base status [1, 15, 23]. Even in healthy fetuses or minimal placental dysfunction with NICHD category I, prolonged labor produces repeated prolonged hypoxic insults during the intrapartum period [22]. Prolonged labor may also induce maternal complications, such as dehydration or uterine rupture, and higher chance of fetal distress. Earlier studies reported that prolonged duration of the second stage of labor was significantly associated with an increase in maternal-fetal lactate concentration [24, 25]. Therefore, pregnant women who have prolonged labor, especially in the second stage of labor, may be at risk of birth asphyxia, even though the FHR tracings show a normal pattern.

Our study found that, when presence of meconium-stained amniotic fluid was combined with NICHD category II, the predictive ability was better than using NICHD category II alone. Previous studies reported that meconium-stained amniotic fluid was a significant risk factor associated with fetal acidemia and/or birth asphyxia [12, 26, 27]. Combining meconium-stained amniotic fluid with NICHD category II has also been found to have a strong association with neonatal morbidities [12, 26, 27]. One important way to explain this finding is that meconium can lead to lung dysfunction due to mechanical obstruction and deactivated surfactants. Meconium also activates systemic immune responses which lead to pulmonary and

systemic inflammation resulting in perinatal asphyxia [28, 29]. Therefore, our results also give credence to the argument that intrapartum FHR tracings alone are not able to predict all cases of perinatal acidemia [5–8, 13, 30]. Therefore, we hypothesized that FHR tracings combined with specific risk factors of individual patients could improve the detection of acidemia, which would reduce the risk of asphyxiation.

The FRI was developed for intrapartum monitoring to more accurately identify hypoxic fetuses [9, 13, 14, 31]. However, no such correlation was found in our study. The main reason was that possibly all of the risk factors of the FRI were not suitable to predict fetuses at risk of perinatal acidemia in our population. In addition, the results indicated that the FRI cannot be applied in all populations. Furthermore, the FRI requires trained interpreters, which is the most crucial limitation for application in general obstetrics.

Our study demonstrated that prediction of perinatal acidemia can be improved when combining risk factors with intrapartum FHR tracings. Our study could identify significant risk factors, which combined with NICHD 3-tier categories I or II improved detection of perinatal acidemia. The sample size was adequate to demonstrate significant risk factors, however, the study was conducted at a single tertiary center, which might reduce the generalizability of the findings. A future prospective studies will need to include a larger sample size for better reproducibility in identifying clinical risk factors, and include more abnormal FHR tracings, especially NICHD 3-tier category III cases.

## Conclusions

In summary, intrapartum EFM according to the NICHD 3-tier FHR classification system combined with selected clinical risk factors can improve the predictive ability for perinatal acidemia in neonates, and it is easy to use in general practice.

## Acknowledgments

The authors would like to thank Miss Walailuk Jitpiboon from the Epidemiology Unit, Faculty of Medicine, Prince of Songkla University for her assistance in data analysis. We also would like to thank Mr. Dave Patterson from the International Affair, Faculty of Medicine, Prince of Songkla University for English language editing.

## Author Contributions

**Conceptualization:** Ninlapa Pruksanusak, Natthicha Chainarong.

**Data curation:** Natthicha Chainarong, Siriwan Boripan.

**Formal analysis:** Ninlapa Pruksanusak, Natthicha Chainarong, Siriwan Boripan, Alan Geater.

**Funding acquisition:** Natthicha Chainarong, Siriwan Boripan.

**Investigation:** Natthicha Chainarong.

**Methodology:** Natthicha Chainarong.

**Project administration:** Natthicha Chainarong, Siriwan Boripan.

**Resources:** Natthicha Chainarong, Siriwan Boripan.

**Software:** Alan Geater.

**Supervision:** Ninlapa Pruksanusak, Natthicha Chainarong, Alan Geater.

**Validation:** Ninlapa Pruksanusak, Natthicha Chainarong, Alan Geater.

**Visualization:** Ninlapa Pruksanusak, Natthicha Chainarong, Alan Geater.

**Writing – original draft:** Natthicha Chainarong, Siriwan Boripan.

**Writing – review & editing:** Ninlapa Pruksanusak, Natthicha Chainarong.

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
