## [Decision Letter · Decision Letter 0]

8 Jun 2022

PONE-D-22-08889Comparison of the predictive ability for perinatal asphyxia in neonates between the NICHD 3-tier FHR system combined with clinical risk factors and the fetal reserve indexPLOS ONE

Dear Dr. Chainarong,

Thank you for submitting your manuscript to PLOS ONE. After careful consideration, we feel that it has merit but does not fully meet PLOS ONE’s publication criteria as it currently stands. Therefore, we invite you to submit a revised version of the manuscript that addresses the points raised during the review process.

We look forward to receiving your revised manuscript.

Kind regards,

Martin Mueller, M.D., PhD

Academic Editor

PLOS ONE

Journal Requirements:

2. Please provide additional details regarding participant consent. In the Methods section, please ensure that you have specified (1) whether consent was informed and (2) what type you obtained (for instance, written or verbal). If your study included minors, state whether you obtained consent from parents or guardians. If the need for consent was waived by the ethics committee, please include this information

Reviewers' comments:

Reviewer's Responses to Questions

**Comments to the Author**

1. Is the manuscript technically sound, and do the data support the conclusions?

Reviewer #1: Partly

Reviewer #2: Yes

2. Has the statistical analysis been performed appropriately and rigorously? 

Reviewer #1: Yes

Reviewer #2: I Don't Know

3. Have the authors made all data underlying the findings in their manuscript fully available?

Reviewer #1: No

Reviewer #2: Yes

4. Is the manuscript presented in an intelligible fashion and written in standard English?

Reviewer #1: Yes

Reviewer #2: Yes

5. Review Comments to the Author

Reviewer #1: The study reports the prediction of asphyxia (?) using a combination of FHR evaluation and risk factors. The approach proposed try to objectify the well-known clinical effort of consider multiple factors and not only the fetal heart rate during labor to bring decision on delivery.

Some methodical limits should be considered. The number of cases is small.

The choice of the sample is not clear. The authors should define or justify indication for continuous fetal heart rate monitoring, UA pH determination, choice of cut-off for asphyxia.

Is the sample a cohort or a convenience one? More information on the suggested or adopted guidelines for FHR interpretation in the center of the study, used to decide the timing of delivery, should be clarified. Also the duration of first and second stage should be eventually evaluated in association to the use of oxytocin.

I noted with interest a higher number of cases of prolonged deceleration in control group, please specify if they were in the context of epidural and eventually report the rate of epidural in the two groups.

Reviewer #2: Thank you very much for the opportunity to review this manuscript. The manuscript covers an important topic and is well written and structured. My comments and suggestions are as follows:

M + M:

1) The term “perinatal asphyxia” is very vague and should be avoided in favor of Hypoxic-Ischemic Encephalopathy (HIE)

2) Why did the authors exclude infants with no umbilical cord gas results? The diagnosis of “perinatal asphyxia” may be made even in their absence.

3) The definition of “perinatal asphyxia” used in the manuscript is not congruent with the reference given (15) and, in particular, does not reflect current clinical practice. The definition of “perinatal asphyxia” should be revised as it requires biochemical criteria (usually a pH of less than 7.00 in the first hour of life) and neurologic criteria (usually encephalopathy or seizures). Infants without neurological abnormalities and a cord blood pH anywhere < 7.20 would be classified as having perinatal acidosis.

Discussion:

1) How did the authors determine their sample size is “large enough”?

2) Why do the authors feel that single center design is a strength?

3) The limitations of this study should be elaborated in more detail, e.g. regarding generalizability.

6. PLOS authors have the option to publish the peer review history of their article (what does this mean?). If published, this will include your full peer review and any attached files.

Reviewer #1: **Yes: **Anna Locatelli

Reviewer #2: No

---

## [Author Response · Author response to Decision Letter 0]

13 Jul 2022

Thank you for valuable comments of both reviewers.

---

## [Decision Letter · Decision Letter 1]

2 Aug 2022

PONE-D-22-08889R1Comparison of the predictive ability for perinatal acidemia in neonates between the NICHD 3-tier FHR system combined with clinical risk factors and the fetal reserve indexPLOS ONE

Dear Dr. Chainarong,

Thank you for submitting your manuscript to PLOS ONE. After careful consideration, we feel that it has merit but does not fully meet PLOS ONE’s publication criteria as it currently stands. Therefore, we invite you to submit a revised version of the manuscript that addresses the points raised during the review process.

We look forward to receiving your revised manuscript.

Kind regards,

Martin Mueller, M.D., PhD

Academic Editor

PLOS ONE

Journal Requirements:

Additional Editor Comments (if provided):

Dear Authors

Thank you for submitting the revised version of the manuscript.

You have addressed the comments of the reviewer 2 sufficiently. Reviewer 1 asked for additional clarifications and kindly ask you to address those as well.

Best regards

Reviewers' comments:

Reviewer's Responses to Questions

**Comments to the Author**

1. If the authors have adequately addressed your comments raised in a previous round of review and you feel that this manuscript is now acceptable for publication, you may indicate that here to bypass the “Comments to the Author” section, enter your conflict of interest statement in the “Confidential to Editor” section, and submit your "Accept" recommendation.

Reviewer #1: (No Response)

Reviewer #2: All comments have been addressed

2. Is the manuscript technically sound, and do the data support the conclusions?

Reviewer #1: Partly

Reviewer #2: Yes

3. Has the statistical analysis been performed appropriately and rigorously? 

Reviewer #1: Yes

Reviewer #2: Yes

4. Have the authors made all data underlying the findings in their manuscript fully available?

Reviewer #1: No

Reviewer #2: Yes

5. Is the manuscript presented in an intelligible fashion and written in standard English?

Reviewer #1: Yes

Reviewer #2: Yes

6. Review Comments to the Author

Reviewer #1: Some points need attention

Methods : definition of pathological deceleration

Number of contraction 8 in 15 minutes to define tachysystole? the definition is different from NICHD classification

Answers to my questions 4 and 5

Why was oxytocin used in all cases if arrest disorders were diagnosed only in a small part of the series? Does the authors refer to oxytocin for other reasons? or oxytocin is routinely used during labor? in both cases specify . The information is relevant for the readers

no cases with epidural? also this information is very relevant and should be added in the results section because the effect of epidural on fetal wellbeing is controversial

table

We need more data on the composition of the samples

mean or median pH and BE of both groups and how many cases were defined as acidemic for pH , for BE or for both

We suggest the authors to perform a similar analysis eventually including a more "stringent" acidemic group

Reviewer #2: The authors have carefully revised the manuscript and have addressed all my comments. I have no further comments.

7. PLOS authors have the option to publish the peer review history of their article (what does this mean?). If published, this will include your full peer review and any attached files.

Reviewer #1: **Yes: **Anna Locatelli

Reviewer #2: No

---

## [Author Response · Author response to Decision Letter 1]

10 Aug 2022

Thank you for your valuable comments

---

## [Decision Letter · Decision Letter 2]

2 Sep 2022

PONE-D-22-08889R2Comparison of the predictive ability for perinatal acidemia in neonates between the NICHD 3-tier FHR system combined with clinical risk factors and the fetal reserve indexPLOS ONE

Dear Dr. Chainarong,

Thank you for submitting your manuscript to PLOS ONE. After careful consideration, we feel that it has merit but does not fully meet PLOS ONE’s publication criteria as it currently stands. Therefore, we invite you to submit a revised version of the manuscript that addresses the points raised during the review process.

We look forward to receiving your revised manuscript.

Kind regards,

Martin Mueller, M.D., PhD

Academic Editor

PLOS ONE

Journal Requirements:

Reviewers' comments:

Reviewer's Responses to Questions

**Comments to the Author**

1. If the authors have adequately addressed your comments raised in a previous round of review and you feel that this manuscript is now acceptable for publication, you may indicate that here to bypass the “Comments to the Author” section, enter your conflict of interest statement in the “Confidential to Editor” section, and submit your "Accept" recommendation.

Reviewer #1: (No Response)

2. Is the manuscript technically sound, and do the data support the conclusions?

Reviewer #1: Yes

3. Has the statistical analysis been performed appropriately and rigorously? 

Reviewer #1: Yes

4. Have the authors made all data underlying the findings in their manuscript fully available?

Reviewer #1: Yes

5. Is the manuscript presented in an intelligible fashion and written in standard English?

Reviewer #1: Yes

6. Review Comments to the Author

Reviewer #1: please verify the definition of pathological deceleration. You stated that the definition included all variable deceleration but this seems not clinically correct and in fact in the table the % of variable deceleration was higher than the % of pathological deceleration. Be sure to be consistent and clinically convinced of your conclusions

7. PLOS authors have the option to publish the peer review history of their article (what does this mean?). If published, this will include your full peer review and any attached files.

Reviewer #1: **Yes: **Anna Locatelli

---

## [Author Response · Author response to Decision Letter 2]

20 Sep 2022

Thank you for your valuable advice.

---

## [Decision Letter · Decision Letter 3]

7 Oct 2022

Comparison of the predictive ability for perinatal acidemia in neonates between the NICHD 3-tier FHR system combined with clinical risk factors and the fetal reserve index

PONE-D-22-08889R3

Dear Dr. Chainarong,

We’re pleased to inform you that your manuscript has been judged scientifically suitable for publication and will be formally accepted for publication once it meets all outstanding technical requirements.

Kind regards,

Martin Mueller, M.D., PhD

Academic Editor

PLOS ONE

Additional Editor Comments (optional):

Reviewers' comments:

Reviewer's Responses to Questions

**Comments to the Author**

1. If the authors have adequately addressed your comments raised in a previous round of review and you feel that this manuscript is now acceptable for publication, you may indicate that here to bypass the “Comments to the Author” section, enter your conflict of interest statement in the “Confidential to Editor” section, and submit your "Accept" recommendation.

Reviewer #1: All comments have been addressed

2. Is the manuscript technically sound, and do the data support the conclusions?

Reviewer #1: Yes

3. Has the statistical analysis been performed appropriately and rigorously? 

Reviewer #1: Yes

4. Have the authors made all data underlying the findings in their manuscript fully available?

Reviewer #1: Yes

5. Is the manuscript presented in an intelligible fashion and written in standard English?

Reviewer #1: Yes

6. Review Comments to the Author

Reviewer #1: .........................................................................................................................................no need of other comments

7. PLOS authors have the option to publish the peer review history of their article (what does this mean?). If published, this will include your full peer review and any attached files.

Reviewer #1: **Yes: **Anna Locatelli

---

## [Editor Report · Acceptance letter]

12 Oct 2022

PONE-D-22-08889R3 

Comparison of the predictive ability for perinatal acidemia in neonates between the NICHD 3-tier FHR system combined with clinical risk factors and the fetal reserve index 

Dear Dr. Chainarong:

I'm pleased to inform you that your manuscript has been deemed suitable for publication in PLOS ONE. Congratulations! Your manuscript is now with our production department. 

Kind regards, 

on behalf of

Dr. Martin Mueller 

Academic Editor

PLOS ONE